# Natural Language Processing for Breast Imaging: A Systematic Review

**DOI:** 10.3390/diagnostics13081420

**Published:** 2023-04-14

**Authors:** Kareem Mahmoud Diab, Jamie Deng, Yusen Wu, Yelena Yesha, Fernando Collado-Mesa, Phuong Nguyen

**Affiliations:** 1Institute for Data Science and Computing, University of Miami, Miami, FL 33146, USA; yxw1259@miami.edu (Y.W.); yxy806@miami.edu (Y.Y.); pxn208@miami.edu (P.N.); 2Department of Computer Science, University of Miami, Miami, FL 33146, USA; jxd3987@miami.edu; 3Department of Radiology, Miller School of Medicine, University of Miami, Miami, FL 33146, USA; fcollado@med.miami.edu; 4OpenKnect Inc., Halethorpe, MD 21227, USA

**Keywords:** Natural Language Processing, breast imaging, breast cancer, radiology reports, pathology reports, systematic review

## Abstract

Natural Language Processing (NLP) has gained prominence in diagnostic radiology, offering a promising tool for improving breast imaging triage, diagnosis, lesion characterization, and treatment management in breast cancer and other breast diseases. This review provides a comprehensive overview of recent advances in NLP for breast imaging, covering the main techniques and applications in this field. Specifically, we discuss various NLP methods used to extract relevant information from clinical notes, radiology reports, and pathology reports and their potential impact on the accuracy and efficiency of breast imaging. In addition, we reviewed the state-of-the-art in NLP-based decision support systems for breast imaging, highlighting the challenges and opportunities of NLP applications for breast imaging in the future. Overall, this review underscores the potential of NLP in enhancing breast imaging care and offers insights for clinicians and researchers interested in this exciting and rapidly evolving field.

## 1. Introduction

Artificial intelligence (AI)- and machine learning (ML)-based technologies, such as Natural Language Processing (NLP), can transform healthcare by deriving new and essential insights from the vast amount of data generated during healthcare delivery. In particular, patient information and literature content in free text form has been growing exponentially to an amount which makes it difficult for health care providers to find and extract meaningful actionable information from [1]. Being able to accurately and quickly identify information stored in free text, such as radiology reports, clinical notes, pathology reports, summary discharge, and others, has the potential to reduce manual workloads, support clinicians in their decision-making processes, triage patients to receive urgent care, or identify patients for research purposes such as clinical trials.

NLP enables a computer system to understand and comprehend information the same way humans do. It helps the computer system understand the literal meaning and recognize the sentiments, tone, opinions, thoughts, and other components that construct a proper conversation. It has been widely applied in industry and business for email filtering, language translation, smart assistants, document analysis, online search, chatbots, social media, and more. Technically, deep learning (DL)-based NLP allows the model to “learn” and “predict” the meaning of human language by training complex DL models using many annotated examples. This method has facilitated significant growth in NLP development, leading to its widespread application and research in various fields, including medicine and industries.

NLP technologies offer promising solutions for assisting physicians, including radiologists, in performing various clinical tasks [1,2,3,4]. It has the ability to greatly impact decision support and utilization by guiding physicians towards optimal workups based on the vast amount of information contained within medical records, including specific clinical circumstances such as risk factors.

Obtaining a comprehensive understanding of the recent advancements in NLP applied to breast imaging is crucial for researchers and developers in the field. By gaining insight into the latest developments, they can broaden their perspective and acquire knowledge about the techniques and methods that facilitate new research in the field. Unfortunately, there is a very limited literature review available on this topic. Only one study conducted by [4] has systematically assessed and quantified the current state of NLP applications in breast imaging. In their research, the authors identified 15 relevant papers on breast imaging that were published up to October 2019. However, their review did not provide a specific categorization or a detailed examination of the identified studies or highlight the need for further research to provide a more comprehensive understanding of NLP’s application in breast imaging. In addition, many physicians are still unaware of NLP technologies and their capabilities in this field.

The primary objective of this review was to provide a comprehensive analysis of the latest research and development of NLP applied to breast imaging, and to summarize the ways in which NLP has been utilized to enhance this field. By focusing on the period spanning from 2013 to February 2023, this review sought to provide an up-to-date understanding of the advancements made in NLP for breast imaging, including its applications and benefits. Additionally, we reviewed the current state-of-the-art in NLP-based technologies for breast imaging and examined the challenges, interests, and future opportunities of NLP applications in breast imaging.

## 2. Materials and Methods

Breast imaging refers to a range of medical imaging techniques used to visualize and examine the breast tissue for the detection and diagnosis of breast diseases, including breast cancer. The most common breast imaging modalities include mammography, ultrasound, magnetic resonance imaging (MRI), and computed tomography (CT). These breast imaging applications encompass a variety of tasks, including breast cancer risk and tissue density assessment, triage and diagnostic tools, as well as the ongoing management of breast disease.

A search of Google Scholar and PubMed.gov was conducted, resulting in a total of 49 papers. For this systematic review, we established inclusion criteria that required research studies on NLP applications in breast imaging. We excluded publications without full text or those not in English. The review followed the Preferred Reporting Items for Systematic Reviews and Meta-Analyses (PRISMA) guideline [5], and any disagreements in the inclusion process were resolved through consensus discussion between K.M. and P.N. A clinical radiology expert (F.C.) provided a final review. We also conducted a hand-search of citation lists to identify any additional studies that met our inclusion criteria. However, we did not assess the risk of bias in individual studies because there was a lack of relevant quality indicators, and the number of studies was limited. Then, each paper was fully read and examined to complete the more detailed sections regarding each of the main NLP tasks in the field of breast imaging. Each of the papers was divided into one of five tasks that we felt adequately covered the field of NLP in breast imaging based on its clinical purposes. The tasks included cancer screening and diagnosis (13 papers), cancer staging (three papers), recurrence (five papers), information extraction (28 papers), and treatment (six papers). Some papers fell into multiple categories. Figure 1 shows the percentage of studies in each category.

We will summarize the performance results used in the reviewed papers regarding popular performance metrics such as positive predictive value (PPV), recall (sensitivity), specificity, the area under the curve (AUC), and accuracy. Table 1 shows the metrics and their descriptions.

We present a brief overview of our findings for each NLP task in Table 2, and we analyzed the NLP methodologies, tools, data cohorts, performance results, strengths, and limitations in the individual studies. We discuss the strengths and limitations of each study in Table 3, and present the main performance metrics for the majority of NLP models in Table 4.

## 3. Background

NLP techniques have been employed to extract crucial information from radiology reports and assist in the interpretation of mammography, breast ultrasound, and breast MRI studies. Through the conversion of medical texts into structured representations, NLP empowers computers to derive information from natural language input, enabling the automatic identification and extraction of information from medical notes and reports [1,26].

Traditional NLP approaches involve rule-based methods and statistical machine learning methods [27]. However, in recent years, the field of NLP has seen a significant shift towards the use of deep learning methods, which have proven to be highly effective in a range of applications [4,28,29]. In particular, deep neural networks, such as recurrent neural networks (RNNs) and convolutional neural networks (CNNs), have been used to great effect in NLP tasks such as language modeling, machine translation, and sentiment analysis.

The typical NLP pipeline involves four main steps: preprocessing, feature representation, feature extraction and processing, and system tasks (see Figure 2).

Text preprocessing is an important step in NLP that is used to clean and transform unstructured textual data into a format that can be used for further analysis. The pre-processing procedure typically involves a series of steps, including cleaning, normalization, tokenization, stop word removal, and stemming or lemmatization [30].

For example, one study [31] used Natural Language Processing and machine learning to extract important information from radiology reports, including breast imaging reports. The authors employed various pre-processing techniques, such as sentence segmentation, tokenization, part-of-speech tagging, and named entity recognition to extract data related to patient demographics, imaging modality, and imaging findings. Another study [32] explored deep learning techniques for radiology report classification and used pre-processing techniques, including tokenization, stop word removal, and stemming.

In addition to these general pre-processing techniques, there are also specialized techniques that can be applied specifically to radiology reports. For example, one study investigated the use of semantic annotation to identify and extract clinically relevant information from radiology reports [33,34]. The authors used a combination of rule-based and machine learning approaches, including pre-processing techniques, such as named entity recognition and semantic type classification, to identify and extract concepts such as body parts, imaging modalities, and imaging findings.

Feature representation is the most critical step, as it involves transforming the raw input data, such as text or speech, into numerical representations that can be used by machine learning algorithms [35]. Some common methods include bag-of-words representations, TF-IDF, word embeddings, and character-level representations [36]. Word embedding is a technique used to encode the meaning of words into real-valued vectors. This approach has gained popularity in recent years due to its ability to capture the semantic meaning of words. Methods for generating word embeddings include neural networks (e.g., word2vec [37], Bidirectional Encoder Representations from Transformers (BERT) [38]), dimensionality reduction, statistical approaches, and rule-based methods.

Feature extraction and processing. One approach to feature extraction is manual feature selection, which involves handpicking relevant features from the text. However, this approach can be time-consuming and may not capture all the relevant information in the text. Alternatively, traditional methods for Natural Language Processing (NLP) include rule-based approaches or statistical models, and machine learning algorithms can be used for these steps. Rule-based approaches use a set of predefined rules to extract information from the input text, while statistical models employ probabilities and statistical techniques to analyze natural language. Machine learning algorithms learn from data and use that knowledge to classify input text or make predictions. Traditional NLP algorithms have been widely used for many years and have been moderately effective. However, they often require significant manual effort to design, and they can struggle with the nuances and complexities of natural language. For example, rules may not always consider the context of the text, making it difficult for the algorithms to capture the meaning of the text accurately.

Deep Learning in NLP. Neural networks are a type of machine learning algorithm that has become increasingly popular in NLP due to their ability to learn complex, non-linear relationships in text data. This makes them particularly effective for understanding contextual information in unstructured medical reports. By using multiple layers of interconnected nodes, neural networks can extract intricate patterns and relationships from the text data, enabling them to produce highly accurate predictions. One of the main benefits of using neural networks in NLP is their ability to handle large volumes of unstructured data. Medical reports, for example, contain vast amounts of information that can be difficult for traditional NLP techniques to process. However, neural networks can easily analyze and learn from this data, allowing them to identify patterns and relationships that would otherwise be challenging to detect.

Recent advances in NLP have seen the emergence of powerful deep learning approaches such as Transformers and Bidirectional Encoder Representations from Transformers (BERT). The Transformer architecture, which relies on a self-attention mechanism, allows the model to selectively focus on different parts of the input sequence, enabling it to capture contextual information in the text [38]. As a result, Transformers have become highly effective in a wide range of NLP tasks and have gained immense popularity since their inception. BERT, which builds on the Transformer model, introduces bidirectional training to language modeling. This means that the model can learn the context of a word based on the words that come before and after it in a sentence, resulting in a much deeper understanding of language context and flow than traditional single-direction models [39]. To further enhance their performance, researchers have adopted transfer learning, a technique where a pre-trained model is fine-tuned on a specific task. BERT is often pre-trained on large corpora of relevant textual data before being fine-tuned on a specific dataset, such as breast imaging reports. The fine-tuning process allows the model to learn task-specific information while retaining the knowledge learned during pre-training, resulting in highly accurate predictions on the specific task. They have been applied to various tasks in radiology, including image captioning, radiology report generation, and medical image analysis.

System tasks and AI modeling. One of the primary tasks of NLP is information extraction. NLP can be used to extract structured information from unstructured text, such as electronic medical records. This is useful in the healthcare industry, where researchers use NLP methods to extract information from patient records to identify disease patterns and trends.

NLP can also be used for classification tasks, such as automatically classifying medical text into diagnosis classes. This can be useful for disease surveillance and clinical decision support. Clustering is another unsupervised learning approach that involves grouping similar documents or texts together based on their contents.

Named entity recognition (NER) is a task of NLP that involves identifying and classifying named entities in unstructured text. Named entities are objects, people, places, organizations, and other entities that have a name or label. Relation extraction (RE) is another important application of NLP that involves identifying and extracting relationships between entities mentioned in unstructured text. These relationships can be expressed in various forms, such as subject-verb-object triples, entity pairs with a relation label, or more complex structures such as graphs or networks.

## 4. Clinical Applications and NLP Methods in Breast Imaging

In this section, we reviewed several papers in detail based on their clinical purposes, including cancer screening and diagnosis, cancer staging, recurrence, information extraction, and treatment.

### 4.1. Breast Cancer Screening and Diagnoses

Natural Language Processing (NLP) has been able to assist with breast cancer screening and diagnosis in various ways. One area of research involves using NLP to extract information from electronic medical records (EMRs) to identify patients at risk for cancer or to assist with diagnosis. This can include identifying patterns in the text, such as symptoms, risk factors, and family history, as well as extracting specific information, such as laboratory results.

NLP models have successfully been developed to examine free-form text from MRI reports to identify index lesions and their corresponding image features. One study utilized an NLP rule-based approach, including concept matching, negation detection, information extraction of lesions, and imaging features [12]. The NLP model examined 1633 Breast MRI reports from 2014 to 2017 and first extracted nine features from each of the found lesions according to the Breast Imaging Reporting and Data System (BI-RADS) descriptors. The model achieved a recall of 100% and a precision of 99.6% in the correct identification of the index lesion. Additionally, the recall and precision of NLP to correctly extract the imaging features from the index lesions were 91.0% and 92.6%, respectively. However, the study was trained and tested using skewed data, as there were no cases for BI-RADS categories 0 or 1.

In an effort to improve NLP performance on clinical data, one study developed an NLP model with the Medical Language Extraction and Encoding (MedLee) system in order to process mammography reports in search of suspicious findings [13]. In this second study, mammography reports were examined for 160 patients, and the NLP system was able to identify suspicious findings in their reports. Both studies demonstrated that NLP is able to produce accurate results within a time period that would be challenging to match, even for a large group of health professionals. Furthermore, applying deep learning NLP algorithms to whole-slide pathology images can greatly improve diagnostic accuracy and efficiency.

NLP has been trained in multiple languages as well. One study discussed the use of NLP on Chinese pathology reports, while another utilized NLP in Italian [14,15]. NLP models have become highly accurate in interpreting free-form text with limited assistance from manual chart abstraction [16].

### 4.2. Breast Cancer Staging

NLP has been used to extract information from pathology reports to assist with breast cancer staging, including identifying specific information in the text of pathology reports, such as tumor size, number of lymph nodes involved, and the presence of certain histologic features. In one study, a total of 150 pathology reports were evaluated by medical professionals to obtain a standard that the NLP would be compared against. The authors used information extraction (IE) and rule-based NLP to extract numerical parameters such as tumor size, lymph node status, and metastases from the impression session of the pathology report. The American Joint Committee on Cancer (AJCC) references were used to classify each category and then group into stages based on a staging protocol. This automatic staging process demonstrated a precision of 73%, 82% recall, 59% specificity, and 72% accuracy [2]. The low performance of this NLP model was attributed to the presence of certain vital information within the reports that the NLP was not built to consider. According to the authors, “processing these sections in the future would improve the performance of this model”. In another study [7], a total of 465 pathology and clinical reports were collected and used for the NLP model to process. The authors used the NLP named entity recognition (NER) and information extraction to extract anatomic and biologic information from pathology reports and medical records, taking into account more information contained in the reports than the previous study did. The authors used a combination of machine learning and rule-based methods for prognostic breast cancer stage detection following the prognostic staging guidelines of the AJCC 8th guidelines to extract prognostic stage elements. These elements included anatomic stage, tumor grade, estrogen receptor (ER), progesterone receptor (PR), and human epidermal growth factor receptor 2 (HER2), which the first study did not. The machine learning model showed 92% and 82% accuracy for different populations in stage prediction. A comparison of these two studies suggests that greater sample size and more emphasis on processing relevant information within reports can greatly improve NLP’s accuracy and suggests that, eventually, labor-intensive manual extraction of pathology reports and unstructured notes could be eliminated entirely by NLP models [6].

### 4.3. Breast Cancer Recurrence

NLP is being used to analyze radiology reports to assist with the prediction of breast cancer recurrence. This can include identifying specific findings in the text, such as the size and location of a lesion, and determining the likelihood of recurrence both distant and local. One particular study capitalized on the increasing availability of electronic health records (EHRs) to improve the efficiency of manual chart abstraction for breast cancer recurrence [8]. In this study, over 1400 breast cancer patients’ clinical notes from 1995 to 2012 were processed by the NLP system. Breast cancer recurrence was defined as an ipsilateral, regional, or metastatic breast cancer diagnosis during a follow up period starting at 120 days after the primary cancer diagnosis until the date of death, or unenrolment from the healthcare system. Recurrences were confirmed using the NLP Information Extraction Apache Ctakes tool [40] and analysis from pathology reports, or progress notes with clinical confirmation, or radiology report findings. The study was limited to stage I and II breast cancers.

Overall, this NLP system was able to correctly identify 92% of recurrences and was also able to estimate diagnoses dates within 30 days for 88% of the recurrences. The NLP model was able to achieve a specificity of 96% and overlooked only five of sixty-five recurrences, with four being due to unavailable electronic documents. In another study, the OncoSHARE database was used to process data from EMRs for 8956 women diagnosed with breast cancer from 2000 to 2018 [9]. In this study, they took the additional step of creating a comprehensive vocabulary to assist the NLP system by interviewing expert clinicians and processing radiology reports, clinical notes, progress notes, and pathology reports. They also conducted two separate approaches with their NLP system, the first being a rule-based model built from rules in metastatic detection and from the literature; the second was a contemporary neural network model. This study was one of the first NLP neural networks to produce highly accurate predictions. The neural network NLP model outperformed the rule-based model in predicted timing of distant recurrence with 0.83 sensitivity and 0.73 specificity. The rule-based model did not do as well, with a specificity of 0.35 and a sensitivity of 0.88. The NLP neural network model still produced errors in the cases in which it could not capture the surrounding context. In addition, the model was not trained nor tested using multiple institution data. BERT-based NLP models have also shown great promise and have been able to accurately predict recurrence [10]. By combining both unstructured and structured clinical data, an NLP model has been able to accurately identify distant recurrences in breast cancer patients [11]. Weakly supervised models have also reached 0.94 AUROC for the prediction of breast cancer recurrence [11].

### 4.4. Information Extraction

NLP models have demonstrated their efficiency in extracting information from various types of medical texts, such as electronic medical records (EMRs), radiology reports, and pathology reports, in order to assist with breast cancer diagnosis, treatment, and prognosis.

One way NLP extracts information in regard to breast cancer is through the use of named entity recognition (NER) and relation extraction (RE) techniques. These two techniques work hand-in-hand as follows: NER is used to identify specific entities such as symptoms, risk factors, medications, and laboratory results, while RE identifies the relationships between these entities, such as whether a patient has a family history of breast cancer or if a medication is being used to treat breast cancer [17].

Machine learning (ML) algorithms are also being trained on large amounts of medical text to identify patterns and connections that may indicate the presence of breast cancer, such as specific symptoms or laboratory results. These patterns can then be used to identify at-risk patients and assist with diagnosis [19].

In order to populate breast cancer screening registries with information extracted from radiology reports, the authors evaluated the accuracy of an automated rule-based approach for identifying imaging findings presented in unstructured text [41]. They examined 139,953 reports, including mammograms, digital breast tomosynthesis, ultrasound, and magnetic resonance imaging (MRI), and compared the extracted data to a “gold standard” established through manual review. The study found that the precision of identifying suspicious calcifications, masses, and distortion (on mammogram and tomosynthesis), masses, cysts, non-mass enhancement, and enhancing foci (on MRI), and masses and cysts (on ultrasound) ranged from 0.8 to 1.0 for recall, precision, and F1 measure.

Using NLP, another study extracted various factors associated with optimal follow-up in women who received breast imaging (ultrasound and mammography) and had three BI-RADS breast findings, such as patient age, marital status, ethnicity, insurance coverage, and income [42]. The authors then utilized univariate analysis and multivariable logistic regression to identify which features were correlated with optimal follow-up. The study included a total of 64,771 unique patients, of which only 2967 (4.6%) had three BI-RADS findings, 74% of optimal follow up.

The authors of another study utilized a convolutional and recurrent neural network model to effectively categorize mammography report text into 33 distinct categories based on individual words [43]. This study found that the neural network classifier outperformed the rule-based method in terms of accuracy. Specifically, the neural network model demonstrated significantly higher keyword accuracy (95.5%) compared to the rule-based method (80.9%). Additionally, the neural network exhibited higher global accuracy (88.3%) than the rule-based method (57.0%).

Recently, BERT-based NLP models have shown great promise for breast cancer information extraction from clinical data. BERT NLP models undergo unsupervised pre-training on large amounts of unlabeled data and can then be fine-tuned to specific breast cancer tasks. This particular form of pre-training gives BERT models a few key advantages over regular NLP models. For example, BERT models are able to interpret longer text inputs and more complex sentence structures, and they are also able to capture more nuanced relationships between words and phrases. These advantages are examined further in the field of breast cancer within several studies [18,44]. This CancerBERT model is portrayed in Figure 3.

Another particular study emphasized the strengths of BERT NLP models [18]. In this study, a pre-trained contextual embedding BERT model was used to separate sessions of breast radiology reports into Breast Imaging Reporting and Data System (BI-RADS) lexicon outlines, achieving 98% accuracy. Then, the authors showed that using both the BERT model pre-trained on breast radiology reports, combined with separated sections segmentation, improved downstream tasks such as field and feature extraction by 17%. Figure 4 shows the example of using the system to extract fields and their information. The dataset used for this research was large with a total of 27,307 patients and 179,396 exams, with exam dates ranging from 2005 to 2020. However, this was a single-institution study.

Most NLP models in the field of breast cancer have been developed with one specific topic area in mind. In one study, however, pre-trained BERT models were used to develop an NLP that can extract features encompassing most clinical areas relevant to breast cancer [45]. These include clinicopathological data, treatment information, prognosis information, and genotype and phenotype information. This study examined pathology reports, clinical encounter notes, radiology notes, surgical operation records, progress notes, and discharge summaries from 100 breast cancer patients. The NLP system consisted of an NER and an RE component and was able to achieve F1 scores of 93.53% for the NER and 96.73% for the relation extraction (RE).

A general purpose NLP system for any particular breast cancer frame would be valuable [46]. One study implemented this comprehensive model to target all clinical information for patients, achieving F1 scores of 93.53% for NER and 96.73% for RE [45]. BERT NLP models outperformed all other machine learning models in extracting phenotype information [44]. Overall, agreement between NLP models and the gold standard for information extraction is high [47]. NLP models have shown promise as a solution for an efficient scalable automation of information extraction from health records [48].

### 4.5. Breast Cancer Treatment

NLP has shown great promise in the area of breast cancer treatment in recent years. The speed at which an NLP model can process large amounts of data far exceeds human capabilities. With the ability to examine more data efficiently, NLP is able to identify patterns and collect insights at a rate much faster than previously possible. In one study that deals with breast cancer treatment, NLP was used to analyze free form text from medical reports and develop predictive models for early and late progression to first-line treatment [20]. The best predictive model for early progression was able to achieve an area under the curve (AUC) of 0.758 using the NLP free-form text approach. The study included approximately 600 patients. This study also developed predictive models using a classic machine learning approach with a dataset of manually extracted features from patients. This other predictive model achieved an AUC of 0.734.

In addition to developing prediction models for breast cancer, NLP has also been used to extract “treatment discontinuation rationale from unstructured EMR notes” to estimate progression-free survival and toxicity incidence [21]. It is important to note that these endpoints are not routinely encoded into electronic medical records. This NLP system was tested and trained on 6115 patients with early-stage breast cancer and 701 patients with metastatic breast cancer. Each group of patients’ data was divided into training (70%), testing (15%), and validation (15%) sub-groups. The results of this study demonstrated that NLP models are indeed capable of extracting a treatment discontinuation rationale with only “minimal manual labeling”. The best logistical regression models identified progression events in metastatic patients with an AUC of 0.752 ± 0.027 and toxicity events in early-stage patients with an area under the curve (AUC) of 0.857 ± 0.014. According to the study, the performance of the NLP model was not significantly different.

NLP models have been implemented on free-text pathology reports to assist with timely follow-ups for abnormal results [22]. With the exponentially growing medical literature on breast cancer, the use of NLP becomes more essential [49].

Table 3 outlines the strengths and limitations of studies within each category. In Table 4, the primary performance metrics for the majority of NLP studies are presented, including the number of samples used and reported performance metrics. By collecting this information from various studies, Table 4 provides an overview of the current results of the studies. For screening and diagnosis tasks, the performance of applications is generally very high, with precision levels exceeding 92.6%. However, it is worth noting that the sample sizes for these studies are limited, with fewer than a thousand samples used. In a study that applied NLP to identify patients who had experienced recurrence using BERT-based modeling, a remarkably high AUC of 0.9883 was achieved with a study sample size of 112,285 patients. Nevertheless, more studies are needed to stage cancers or assist in treatment plans using NLP with larger sample sizes.

## 5. Discussion

In this section, we discuss the existing limitations and challenges of NLP applications for breast imaging. We also discuss open research questions, clinical interests, and some recommendations that may benefit this field in the near future.

### 5.1. NLP Applications in Breast Imaging

Manually labeling large data cohorts is impractical; hence, NLP tools allow researchers to build data cohorts by automatically extracting outcomes that serve as labels for training deep learning models at scale [50]. This has the potential to accelerate the discovery of predictive biomarkers for prognosis, therapeutic response, or the risk of adverse events. However, most studies reviewed in this paper are still simple proof-of-concept results, and mature NLP DL-based systems have not yet been trialed in clinical settings. Additionally, FDA-cleared NLP software has not been deployed in medical devices as a stand-alone application.

While traditional rule-based methods are still valid and easier to interpret, advanced deep learning methods have been shown to outperform them. To improve the robustness and interpretability of NLP systems, a combination of these methods (hybrid methods) can be used [51]. A study found that hybrid methods that combined deep learning and rules improved performance for the diagnostic surveillance category.

The size of the dataset used to draw experimental conclusions is crucial, and accurately reporting these measures is essential for reproducibility and comparability in future studies. Most studies reviewed in this paper had sample sizes of up to thousands, except for NLP studies used to label samples for training AI applications. To address this, several studies have explored combining word embeddings and ontologies to create domain-specific mappings, which could avoid the need for large amounts of annotated data [52,53]. In addition to reducing data requirements, such combinations could improve coverage and performance compared to more conventional techniques for concept normalization.

Most NLP studies reviewed in this paper used retrospective data from a single institution, leading to AI models that do not perform well in new settings, a phenomenon known as domain shift or data drift.

For instance, AI models that rely on BI-RADS lexicon descriptors and categories as guided features for extracting information from breast imaging reports may not work when applied to institutions that do not follow BI-RADS references or use different versions of the lexicons [18]. To address this issue, larger training datasets from multiple institutions can be used to improve the generalizability of NLP systems.

In addition, the performance of NLP systems is negatively impacted by the ambiguity in natural languages such as lexical ambiguity in phrase variations or syntactic ambiguity. These issues can be improved by using a hybrid combination of DL/NLP with traditional methods such as word sense disambiguation, part-of-the-speed, semantic, ontology taggers, and rule-based approaches.

External validation of NLP research and development for breast imaging was extremely low in the reviewed studies. This may be due to constraints in terms of resources and a lack of external datasets. There are much-needed common datasets and benchmarks for developing and validating algorithms. In addition, more evidence is needed for comparison between NLP systems and human performance (human observer) before they can be used in clinical practice. The robustness of the system can be improved by well-designed training datasets and algorithms, careful validation, external validation, and human observer comparison. A practical step-by-step guide to the development and deployment of AI applications in clinical settings is presented in this paper [54].

NLP approaches using Bi-directional Encoder Representations from Transformers (BERT)-based embedding models and its pre-trained models and embeddings are becoming popular, among other reasons, due to it supporting better contextual representation. Although the pre-trained models often require fine tuning, this can reduce computational cost and improve generalization. However, the approaches based on BERT embeddings will need to be carefully evaluated since there is some evidence that BERT-based embedding failed to capture negation and context in radiology reports [55,56]. Recent evidence shows that embeddings generated by BERT fail to show generalizable understanding of negation [57], an essential factor in interpreting radiology reports effectively.

### 5.2. Open Research and Clinician Interests

In this session, we discuss use cases and clinical interests associated with NLP which potentially benefit breast imaging.

#### 5.2.1. Clinical Decision Support

By extracting relevant historical patient data, current clinical situation, reason for exam, and best practice standards—such as the American College of Radiology (ACR) Appropriateness Criteria—NLP could determine whether the exam should be a routine screening or a diagnostic exam, as well as the most appropriate imaging modality to use. This can potentially support health care providers, radiologists, and radiology technologists and guide them towards optimal exam and scanning protocols, enhancing standardization, efficiency, and quality while reducing patient risks. Additionally, NLP-enabled outcomes could serve as clinical decision-support tools that evaluate patient risks and offer prognostic predictions.

#### 5.2.2. Computer Assist Coding

Assisting radiologists in creating diagnoses and ICD-10 codes in the impression session, together with the list of findings, is relevant to the sustainability of radiology practices, and an NLP system can fulfill this need [58]. The NLP system can analyze what the radiologist has included in the report and provide suggestions for improvements such as adding more impressions, diagnoses, and the correct ICD-10 codes. This approach can potentially streamline the coding process, improve radiologists’ efficiency, and maximize claims.

#### 5.2.3. Computer Assist Reporting

Integrating NLP with AI in breast imaging can help highlight important findings and suggest text to add to reports based on knowledge from large report bodies, the literature, and compliance guidelines. Digital data from standard mammographic and tomosynthesis breast exams can be transmitted to a server accessible to an AI/NLP system which analyzes the data, generating breast density information for the interpreting radiologist while automatically populating the appropriate information using NLP tools into the radiologist’s report. Such a system holds the potential to improve radiologists’ workflows and efficiency [58].

#### 5.2.4. Improved Classification of High Risk Breast Lesions

By integrating quantitative imaging features of original biopsied micro calcifications in diagnostic imaging with deep learning, as well as text features of pathology reports with NLP tools and/or histopathologic slide imaging features, alongside patient risk factors and characteristics, AI can enhance the accuracy of predicting malignancy in high-risk lesions. This approach offers a more dependable and reproducible method for multimodal evaluation, potentially decreasing the rate of unnecessary surgical excision [58].

#### 5.2.5. Lesion Detection and Classification

By employing a combination of AI/NLP tools, radiologists can accurately pinpoint a lesion of interest and obtain a list of similar-appearing lesions with biopsy-proven pathologic outcomes extracted by NLP tools. This approach can bolster confidence in management decisions, reducing the need for unnecessary recall imaging and biopsies [58]. In the case of lung nodules, a hybrid model incorporating deep learning computer vision and CT-report NLP demonstrated the ability to identify nodules that would have been missed by text-only identification, therefore avoiding additional false positives [59]. AI/NLP algorithms can potentially provide radiologists with automated continuous numerical risk scores for malignancy based on radiologic lesion morphology, NLP-extracted features from pathology results and/or actual histopathologic slide imaging features, and patient characteristics such as age, family history, breast density, prior cancer, and mutations. This approach can guide the best clinical recommendation. Such systems can update in real-time as patients are treated and monitored over time, offering an evolving picture of the patient’s prognosis.

## 6. Conclusions

This review provided an overview of the recent advances in NLP for breast imaging based on 49 papers published during the period 2013–February 2023. A significant 28 studies concentrated on the extraction of key information related to breast lesions, including their type and characteristics, followed by 13 studies focused on NLP for cancer screening and diagnosis. We have summarized different NLP methods used to extract relevant information from clinical notes, radiology reports, and pathology reports, and how this information can be used to improve the accuracy and efficiency of breast imaging. Our analysis reflected the recent trend in the NLP research field, which has shifted towards Bi-directional Encoder Representations from Transformers (BERT)-based embedding models. BERT-generated embeddings are known to deliver better contextual representations and improve task performance. However, more exploration is needed to weigh the performance gains against the benefits of generalizability for breast imaging text. Additionally, we have discussed the challenges and opportunities of NLP applications for breast imaging in the future. Overall, this review highlights the potential of NLP for improving breast imaging care and provides insights for researchers and clinicians interested in this exciting and rapidly evolving field.

## Figures and Tables

**Figure 1 diagnostics-13-01420-f001:**
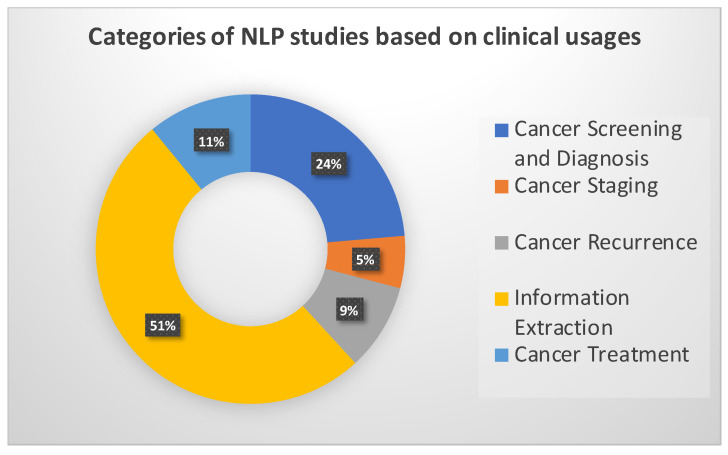
The percentages of studies in each category based on clinical usage.

**Figure 2 diagnostics-13-01420-f002:**
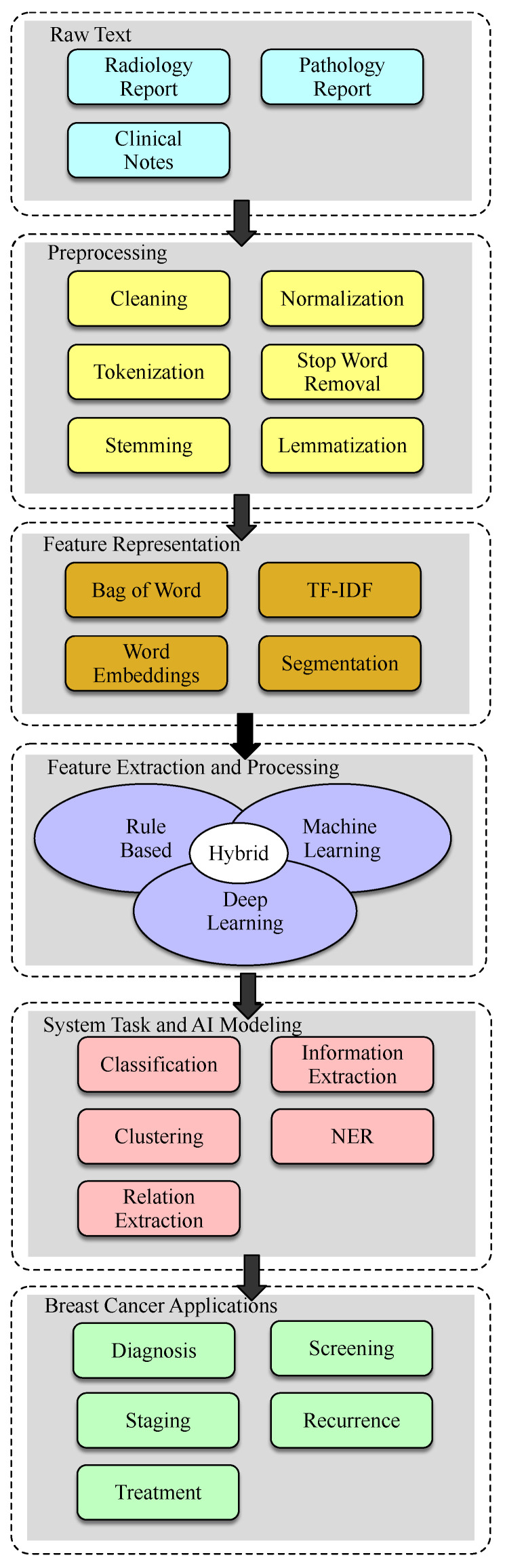
The typical NLP pipeline involves four main steps: text preprocessing, feature representation, feature extraction and processing, and system tasks. Raw text is the input layer and breast cancer applications are the last layer. We discuss the detailed applications in Section 4.

**Figure 3 diagnostics-13-01420-f003:**
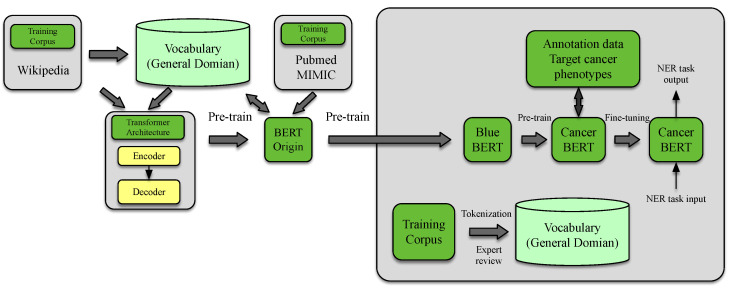
cancerBERT [44] models undergo pre-training on a large corpus of text data such as Wikipedia. Once pre-training is complete, they are then fine-tuned to a specific NLP task where it is trained on a smaller, more specific dataset.

**Figure 4 diagnostics-13-01420-f004:**
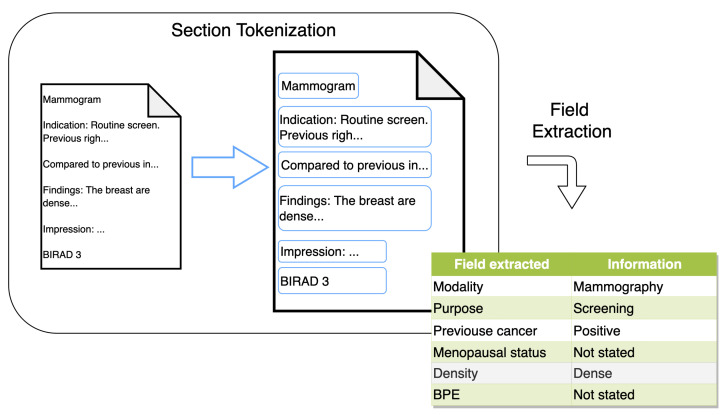
Authors [18] employed a Bi-RADS Bert model to segment the reports prior to extracting the fields to enhance model’s accuracy.

**Table 1 diagnostics-13-01420-t001:** Performance metrics. Precision (Prec). True positive (TP). False positive (FP). True negative (TN). False negative (FN).

Metric	Description	Formula
Accuracy	Correctness on average	(TP + TN)/(TP + FP + TN + FN)
PPV	Positive Predictive Value, Precision	TP/(TP + FP)
Sensitivity	true positive rate, Recall	TP/(TP + FN)
Specificity	true negative rate	TN/(TN + FP)
F score	F1, harmonic mean of precision and recall	2(Prec. × Recall)/(Prec. + Recall)

**Table 2 diagnostics-13-01420-t002:** Category and summary of NLP in breast imaging applications based on clinical usage.

Task, [Papers]	Methods	Summary	Results
Staging [2,6,7]	Information extracted from pathology reports with machine learning and rule-based systems.	Pathology reports are processed with NLP to extract parameters for breast cancer staging, namely tumors, lymph nodes, and metastases.	Results have been promising with multiple NLP models achieving over 90% accuracy in identifying breast cancer staging.
Breast cancer recurrence [8,9,10,11]	NLP with BERT model, data from OncoShare Database	NLP has been used to detect patient-specific timing of metastatic recurrence, and calculate the probability and identify both distant and local recurrences.	Best NLP models were able to identify over 90% percent of recurrences and estimate diagnoses dates for most patients within 30 days.
Screening and Diagnoses [12,13,14,15,16]	Studies examined free-formed text reports and extracted features according to BI-RADS.	NLP has been used to identify index lesions in breast cancer patients and also extract information on them. Manual review was conducted to ensure the accuracy of NLP models.	Identification of index lesion has shown to be extremely accurate, almost 100%
Information Extraction [17,18,19]	Text classification, named entity recognition, sentiment analysis, and concept extraction.	Information extraction with NLP has been used for prognostic stage detection. Able to identify patterns and insights in a short amount of time, performance improves significantly with manual assistance.	NLP systems have been able to accurately extract information with over 90% sensitivity and precision.
Treatment [20,21,22]	Trained and tested on electronic health records of real-world breast-cancer patients.	NLP to develop early predictive models for patient response.	Best predictive models with NLP achieved area under the curve (AUC) of 0.758.

**Table 3 diagnostics-13-01420-t003:** Summary of pros and cons in the reviewed NLP studies.

Task	Pro’s	Con’s
Cancer Screening and Diagnoses	Can standardize and streamline the diagnosis process, reducing human error and improving consistency in interpretation of data. Very accurate in diagnosis and identification of index lesions.	Needs large amounts of labeled data to train the model. Still has the potential for error and can therefore lead to false positives and negatives. Potential to increase the workload for medical professionals who will need to verify the results of the NLP model.
Cancer Staging	NLP Can take into account all of the available data when dealing with breast cancer staging to improve accuracy.	NLP may be unable to fully replicate a clinical examination.
Recurrence	NLP Has proven to be very accurate in predicting breast cancer recurrence.	Key differences from patient to patient can pose challenges for NLP models to adapt to.
Information Extraction	Can save a lot of time by quickly analyzing and interpreting large amounts of clinical data.	Difficulties dealing with different medical terminologies.
Treatment	Can improve the quality and completeness of patient information that is available to physicians and researchers. Can be used to identify patterns and insights from large amounts of data that would have gone unnoticed. Can also be used to monitor online platforms for early detection.	Difficulties in interpreting nuances and context of human language can lead to mistakes and inaccuracies with treatment. Privacy and security concerns.

**Table 4 diagnostics-13-01420-t004:** Performance of NLP studies in different categories and its sample size.

Task	Methods	Accuracy	AUC	Recall	Precision	Sample Size
Staging	Extracting parameters from pathology reports [2]	72	-	82	73	150
	Prognostic stage detection in rural/urban regions [23]	-	93/83	-	-	465
Screening and Diagnosis [12]	Identification of Index Lesions	-	-	100	99.6	478
	Identification of BI-RADS Categories	-	-	96.6	94.8	478
	Extracting Imaging Features	-	-	91	92.6	478
Recurrence	Identifying recurrences [24]	92	-	-	-	1472
	Identifying patients who experienced recurrence (BERT-base) [10]	-	0.9883	-	-	112,285
	Predicting timing of metastatic recurrence [9]	-	-	-	-	894
Information Extraction	IE to Determine Recruit Eligibility for Studies [25]	-	-	-	91.6	-
	Clinical NER/RE using Bi-char-LSTMs and random forest classifiers [17]	-	-	0.82/0.94	0.80/0.82	800
	BI-RADS BERT perform section segmentation and extract information (density, previous cancer) [18]	95.9	-	-	-	155,000
Treatment	Identifying toxicity events in early stage patients [21]	-	0.857	-	-	6115
	NLP free-text to predict early/long progression to first-line treatment [20]	-	0.758/0.752	-	-	610

## Data Availability

Not applicable.

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
