# Peer review of "Natural Language Processing for Breast Imaging: A Systematic Review"

_diagnostics, 2023, doi:10.3390/diagnostics13081420_

Round 1
Reviewer 1 Report
Authors are encouraged to follow the Preferred Reporting Items for Systematic Reviews and Meta-Analyses (PRISMA) guidelines when presenting inclusion exclusion criteria for selecting data source / papers for reviews, and other PRISMA checklist items.
Minor typos:
Line 259 : “processing radiology radiology reports” ïƒ “processing radiology reports”
Line 441 : “Computer assisted coding “ ïƒ remove this
Author Response
We addressed the reviewer comment by following PRISMA guideline and stated inclusion exclusion, bias in session 2. Materials and Methods
And added citation for PRISMA guidelines.
We fixed the typos
Reviewer 2 Report
Dear Authors,
The review paper is excellent, comprehensive, well-written, and well-structured.
I do not have any concerns.
Author Response
No changes.
Reviewer 3 Report
Manuscript title: Natural Language Processing for Breast Imaging: A Systematic Review
Natural Language Processing (NLP) is a promising tool for improving breast imaging in the diagnosis and treatment of breast cancer and other breast diseases. It can extract relevant information from clinical notes, radiology reports, and pathology reports to improve the accuracy and efficiency of breast imaging. NLP-based decision support systems are also being developed for breast imaging. This review highlights the potential of NLP in enhancing breast imaging care and offers insights for clinicians and researchers interested in this field.
This manuscript provides a good overview of the literature from 2013 to the present. The author mentions the limitations and challenges of some of the papers. Considering these challenges and recent advancements, can the author summarize all advancements in one place and provide a pipeline or algorithm for new researchers to follow when considering these steps in pathology, radiology, or clinical reports?
Author Response
To address your comments, we have added some improvements. Firstly, we have included Figure 1 as a visual aid to summarize NLP applications in each category.
In the conclusion section, we have added a sentence highlighting the most studied NLP applications in the field of breast imaging. Specifically, we note that "A significant 28 studies concentrated on the extraction of key information related to breast lesions, including their type and characteristics following by cancer screening and diagnosis (13 studies)."
To provide further context for readers, we have added a description of Tables 3 and 4 at the end of Section 4. This will help readers better understand the strengths and limitations of studies in each category and the primary performance metrics reported in Table 4.